# Preliminary Experience with Electronic Brachytherapy in the Treatment of Locally Advanced Cervical Carcinoma

**DOI:** 10.3390/cancers17142286

**Published:** 2025-07-09

**Authors:** Desislava Hitova-Topkarova, Virginia Payakova, Angel Yordanov, Desislava Kostova-Lefterova, Mirela Ivanova, Ilko Iliev, Marin Valkov, Nikolay Mutkurov, Stoyan Kostov, Elitsa Encheva

**Affiliations:** 1Department of Radiotherapy, UMHAT “Dr. Georgi Stranski”, 8A Georgi Kochev Blvd., 5809 Pleven, Bulgaria; vpayakova@gmail.com (V.P.); dr.ivanova.mirela@gmail.com (M.I.); marinvalkov@abv.bg (M.V.); 2Scientific and Innovative Program Med for Health, Medical University Pleven, 1, Saint Kliment Ohridski Street, 5800 Pleven, Bulgaria; angel.jordanov@gmail.com (A.Y.); ilkoiliev_92@abv.bg (I.I.); 3Department of Gynecologic Oncology, UMHAT “Dr. Georgi Stranski”, 8A Georgi Kochev Blvd., 5809 Pleven, Bulgaria; 4National Cardiology Hospital, 65 Konyovitsa Street, 1309 Sofia, Bulgaria; 5Complex Oncology Centre, 86 Demokratsia Blvd., 8000 Burgas, Bulgaria; n_mut@abv.bg; 6Department of Gynecology, Medical University Varna “Prof. Dr. Paraskev Stoyanov”, 9002 Varna, Bulgaria; drstoqn.kostov@gmail.com; 7Department of Radiotherapy, UMHAT “Saint Marina”, 1 Hristo Smirnenski Blvd., 9010 Varna, Bulgaria; dr.encheva@gmail.com; 8Department of Imaging Diagnostics, Interventional Radiology and Radiotherapy, Medical University of Varna, 9000 Varna, Bulgaria

**Keywords:** cervical cancer, electronic brachytherapy, radiotherapy

## Abstract

The current study aims to investigate the dosimetric and clinical results of the treatment of locally advanced cervical cancer with electronic brachytherapy. A total of 25 patients were treated with chemoradiotherapy and intracavitary electronic brachytherapy. After a median follow-up of 12 months, 84% of the patients achieved local control in the cervix, and a complete response was noted in 72% of the patients. The method demonstrated safety with its low toxicity. This paper discusses methods for further improvement of the treatment protocol and highlights the necessity for more studies on the matter.

## 1. Introduction

Cancer of the uterine cervix is the fourth most common cancer diagnosed in individuals assigned as female at birth, with 662,301 new cases worldwide in 2022. The age-standardized incidence in Bulgaria is 15.4 per 100,000, which is the fourth highest among European countries, and the age-standardized mortality is 6.1 per 100,000 [1]. Poor prevention, as well as inadequate access to screening and treatment, are possible explanations of the higher burden of cervical cancer [2].

The focus of the current treatment guidelines is on improving both oncological outcome and quality of life. A general recommendation is to avoid the combination of radical surgery and adjuvant radiotherapy to reduce treatment-related morbidity. All patients with locally advanced cervical cancer (LACC) are subject to definitive external beam radiotherapy (EBRT) with concomitant chemotherapy and image-guided brachytherapy (IGBT) [3,4]. It is currently recommended to follow the protocol described in the EMBRACE II study [5] and to reduce the overall treatment time, as it could provide better local control [6]. Even though IGBT is irreplaceable in terms of treatment outcomes, it is not widely available in certain geographic regions [7]. Studies by Guedea et al. demonstrated that European countries with fewer resources had fewer brachytherapy (BT) centers [8]. According to another study, the number of patients receiving BT in Bulgaria was also limited due to a lack of enough local resources [9]. A paper by the president of the Guild of Bulgarian Radiotherapists, Professor Hadjieva, demonstrated the tendency of insufficiency in BT units [10]. According to the Directory of Radiotherapy Centers, there were 11 brachytherapy units in Bulgaria in 2024 [11]. However, there are no official data on the availability of cervical IGBT.

In 2015, four electronic brachytherapy (EBT) units were delivered to four university hospitals in Bulgaria, equipped for intraoperative partial breast irradiation. The Elekta Xoft^®^ Axxent^®^ Electronic Brachytherapy (eBx^®^) System^®^ (Elekta, Stockholm, Sweden) uses a 50-KV X-ray source to provide high dose rate (HDR) brachytherapy with various applicators. Following the high demand for brachytherapy of gynecologic cancers, the unit was further equipped for the implementation of the technique. EBT was considered a promising option, due to the nature of the source, which requires minimal shielding, does not produce radioactive waste, and has a reduced risk of incident occurrence [12].

The purpose of the current study is to report the first institutional dosimetric and clinical results from the treatment of LACC with EBT, as well as to use the findings to optimize the treatment protocol.

## 2. Materials and Methods

Patients were enrolled prospectively following the inclusion criteria: age over 18, histologically proven cervical cancer stage IB-IVA, no previous hysterectomy, no previous pelvic or abdominal radiotherapy, and lack of general contraindications for brachytherapy. All patients signed informed consent.

EBRT was delivered as volumetric modulated arc therapy (VMAT) with concomitant boosts in involved regional lymph nodes where applicable. Concurrent chemotherapy was Cisplatin at 40 mg/m^2^ weekly.

A total of 25 patients with a median age of 60 years (range 29–83 years old) were treated with cervical EBT from November 2022 to June 2024. Tumor stage and histology distribution are presented in Table 1.

Tumor response was assessed during the fifth week following EBRT by magnetic resonance tomography (MRT) and gynecological examination. The gynecologic oncologist evaluated the possibility of inserting the tandem in the cervical canal, the appropriate tandem angle, and the size of the ovoids. All eligible patients were scheduled for EBT immediately after the end of the EBRT course, two consecutive days in the first week, and two consecutive days in the second week, for a total of four fractions.

The procedure began with short-term general anesthesia and insertion of a Foley catheter in the urinary bladder, which was installed with from 100 to 200 mL of saline to ensure the proper position of the uterus. After examination under anesthesia and transabdominal or transrectal ultrasonography (Resona R9 platinum edition, Mindray, Shenzhen, China), the radiation oncologist, the gynecological oncologist, and the medical physicist made the final decision on the proper applicator geometry. The set consists of two types of colpostats, three sizes of ovoids, and four angles of tandems (Figure 1). The tandem and colpostats were locked to the base plate by a clamp, and the cervical stopper device ensured stable positioning. A vacuum bag further immobilized both the patient and the base clamp in order to avoid changes in the position of the applicator while being transferred for a computed tomography (CT). The CT images were acquired with a slice thickness of 2 mm. After visual inspection for unfavorable geometry or perforation of the uterus, CT images were transferred to the treatment planning software (TPS), BrachyCare version 1.1.0, based on Task Group No. 43 Report. The defined organs at risk (OARs) were the bladder, rectum, sigmoid colon, and small bowel. The high-risk clinical target volume (CTV HR) was the whole cervix and gross tumor, while the intermediate clinical target volume (CTV IR) was defined by adding a 1 cm margin in the lateral and craniocaudal direction and a 0.5 cm margin in the anteroposterior direction, cropped at the OARs. The prescribed physical dose per fraction to the CTV HR was 700 cGy. The planning aims and constraints per fraction are presented in Table 2. The CTV IR was of the lowest priority, and, when all other targets were achieved, the goal was 100% coverage with 50% of the prescribed dose (350 cGy).

Dosimetry planning included defining the applicator, the prescribed dose per fraction, and the dose prescription point, which was determined as the most distant from the tandem point that lies on the outer border of CTV HR. For optimal dose distribution, the medical physicist optimized the source dwell positions and treatment time in each channel of the applicator. Considering both the coverage of the target area and the doses in OARs, each plan was approved by the medical physicist and the radiation oncologist. Afterwards, the treatment data were transferred to the Xoft Axxent controller system. Before the treatment procedure, the medical physicist completed the steps of the quality assurance program of the system, including source length calibration, verification of the source position in the robotic arm, cooling system function check, and source strength calibration. All printed data from the TPS were double-checked for consistency with the data imported into the controller. The procedure was repeated for each fraction of treatment.

The patients were scheduled for follow-up gynecological examinations on the first, third, sixth, ninth, twelfth, eighteenth, twenty-fourth, and thirty-sixth months after the treatment. MRT was performed one month after the completion of therapy, and a positron emission tomography–computed tomography (PET/CT) was performed after the third month and then annually for five consecutive years. The radiation oncologist and gynecologist evaluated the toxicity on each visit and scored by the Common Terminology Criteria for Adverse Events (CTCAE) version 5.0. Patients were encouraged to report adverse events at any time during the follow-up period.

## 3. Results

All patients completed the treatment successfully with a median overall treatment time (OTT) of 52 days (range 39–89) in accordance with the planned schedule of a total of four fractions of 700 cGy to the CTV HR. The most frequently used tandem angle was 15° (14 cases), followed by 30° (7 cases), while 45° was used in three patients and 0° in one. The median value for the initial CTV HR volume was 15.4 cm^3^ (range 8.9–37.9), and the overall treatment time per session of 7 Gy was 13.5 min (range 9.1–23.6). CTV HR coverage was individually assessed for each fraction of treatment in accordance with the clinical situation. The median value of CTV HR receiving the prescribed dose per fraction was 94% (range 88–97), and the median value of the cumulative dose in EQD_2_ (equivalent dose in 2 Gy fractions) from EBRT and EBT in 90% of the CTV HR was 90 Gy (range 86–93). The mean doses in organs at risk received in the most exposed 2 cm^3^ volume per fraction were: bladder D_2cc_—410 cGy (range 239–739), rectum D_2cc_—189 cGy (39–801), and sigmoid D_2cc_—255 cGy (range 104–703). The cumulative doses in the OARs from both EBRT and EBT in EQD_2_ were calculated with an α/β ratio of 3, while, for the CTV HR, an α/β ratio of 10 was used. The detailed data on dose distributions for each patient are represented in Table 3.

The patients were followed up for a median of 12 months (range 6–24). Early treatment-related toxicity was evaluated during the course of EBT, one month, and three months afterwards. Late toxicity was scored on each follow-up after the sixth month. Table 4 presents the events of persistence and progression. All failures were observed in patients with NKSCC G2, stages IIIB, IIIC1, or IIICr. A total of 72% of the patients achieved a complete response without further treatment, and local control in the cervix was 84%.

Early toxicity among patients without recurrences included one patient with vaginal discharge G1 at the first month, which resolved by the third month. Late toxicity was recorded in the first two patients, who had proctitis G2 and G1, respectively, at the ninth month of follow-up, which was resolved in the course of the next three months.

## 4. Discussion

All of our data results for treatment time duration, CTV HR coverage, doses in OARs, and adverse events are comparable with the available published clinical studies on the treatment of cervical cancer with EBT [13,14].

The first strong clinical evidence of the relationship between the CTV HR coverage and the local recurrence rate was published in 2009 by Dimopoulos et al. [15]. They found that a CTV HR D90 of 87 Gy EQD_2_ provided more than 95% local control, while, in the group of patients who achieved a lower than 87 Gy D90 (median 75 Gy), the rate of failures was 20%. The importance of CTV HR coverage in BT treatment in terms of achieving local control in the cervix was confirmed in the retroEMBRACE study, where patients were treated with MRI-guided BT [6]. In this study, with a median follow-up of 46 months, a D90 of 85 Gy EQD_2_ was associated with a local control rate of 94% for stage II patients and 86% for stage III patients. Our experience with EBT demonstrated that the desired target coverage can be achieved in most cases with a median CTV HR D90 of 90.02 Gy EQD_2_. No isolated local failures have been observed that could confirm that EBT delivered the necessary target coverage to the cervix. Three patients, in whom the necessary D90 was reached, had persistent local disease as well as distant metastases, whereas other major risk factors could contribute to these failures, and further investigation is necessary.

The volume of the CTV HR at the first fraction of brachytherapy is a significant predictive factor for 5-year overall survival (OS), alongside FIGO stage, lymph node positivity, and the delivery of concurrent chemotherapy, according to the first nomogram developed for patients with LACC treated with IGABT [16]. It is notable that our population of patients had a good response to initial chemoradiotherapy, and the CTV HR volumes were favorable. Both the reduction in the tumor volume and its diameter have been proven as predictive factors for improved progression-free survival (PFS) and disease-free survival (DFS) in recent years [17,18,19]. In the current patient cohort, only two patients had a restaging tumor volume over 6.25 cc and a reduction rate of 56% and 63%, respectively. The first one was the patient with the lowest stage in the group and achieved complete remission with no events during a follow-up period of 24 months. The other patient’s tumor, which had an initial volume of 21 cc and involved the uterus, displayed quick regrowth that was evident in the first month after the completion of treatment. This raised concern, and an early PET/CT was performed at the end of the second month, as well as a repeat MRI. The studies showed further tumor growth to the pelvic wall and metastatic lymph nodes in the paraaortic and mediastinal regions. Chemotherapy and a PD-L1 inhibitor were recommended, but she refused treatment and opted for palliative care later, dying 13.5 months after the completion of treatment.

The OTT is an important independent predictive factor for local failure. In a study published in 2015 by Mazeron et al. [20], its significance in the era of chemoradiotherapy and IGABT was confirmed. They established a cut-off of 55 days, beyond which the probability of achieving local control would decrease by 0.63% per day. Another calculation in the article by Tanderup et al. [6] leads to the conclusion that an increase in the OTT by one week results in a decrease in 3-year local control of 1% for patients with a CTV HR volume of 20 cc, and 1.2% for those with a CTV HR volume of 30 cc. In the current study, two patients had an OTT period elongation of 5 weeks (numbers 6 and 7), and two patients had one of two weeks. The first one needed an emergency nephrostomy change and systemic antibiotic treatment due to a urinary tract infection, which led to the prolonged treatment period. In the other case, the patient was referred to another brachytherapy center as she was considered a proper candidate for interstitial needle treatment. After a perforation of the isthmic part of the uterus during the first applicator insertion, she was sent back to our center for treatment. In both patients, a D90% of 90 Gy was achieved. However, considering the high rectal doses and the comorbidities, further escalation was not attempted. The other two patients, who exceeded the proper OTT by two weeks, have had more than a year of follow-up with excellent locoregional control so far.

All seven patients whose disease progressed had lymph node metastases at diagnosis, detected by imaging techniques. No patients had been surgically staged, i.e., none of the metastases had been removed. During EBRT, the involved lymph nodes were treated with 55 Gy in 25 fractions. Two patients experienced in-field nodal failure, and three patients developed lymph node metastases outside the treatment field. According to the analysis of the EMBRACE data [21], the main risk factors for nodal failure are nodal disease, tumor width at diagnosis, local failure, and nodal disease in the common iliac or paraaortic region. This study also showed that prophylactic irradiation of the paraaortic nodes improves the local control in the region. The size of the metastatic lymph nodes has also been disputed as an independent prognostic factor for DFS and overall survival (OS) [22]. In the current patient cohort, five out of seven patients who experienced local failure or distant progression had lymph node metastases with a short axis diameter larger than 15 mm. Even though a simultaneous integrated boost of 55 Gy is considered feasible [23], it is important to note that EBT has a steep gradient dose distribution and does not contribute to nodal dose. Considering not only size but also other negative prognostic factors, like high SUVmax values [24], pretreatment hypoxia [19], and biomarkers [25], it would be reasonable to aim for a further escalation of the dose in the high-risk involved lymph nodes during EBRT.

Another option for treatment intensification to reduce the risk of distant metastases development and improve the treatment outcome is to deliver at least five weekly cycles of concomitant cisplatin [26]. In the current study, six out of seven patients who did not achieve remission received five cycles of cisplatin, and one patient received four due to the low creatinine clearance, neutropenia, and gastrointestinal toxicity. Recently, there has been a breakthrough in the field of achieving a higher PFS period in patients with high-risk LACC by the addition of the immune checkpoint inhibitor pembrolizumab to chemoradiotherapy [17,27]. As this novel treatment is on its way to becoming a global standard of care, it is to be introduced in the current institutional protocol, especially considering that the observed toxicity of chemoradiotherapy and EBT after the first month of follow-up has been low.

Even though target coverage is of the highest priority in dose optimization, it is immediately followed by the hard constraints for the OARs [28], as they are related to acute and late treatment-related toxicity.

The soft constraint for the bladder D_2cc_ is <90 Gy EQD_2_, and the hard constraint is <80 Gy, which was met by 72% of the patients in this study. A dose over 90 Gy is related to a higher probability of bleeding, while doses over 85 Gy are related to pain or difficulty in voiding the bladder, and doses exceeding 80 Gy are a risk factor for cystitis [29]. In this study, no toxicity related to the bladder has yet been recorded. While the ICRU bladder point is a surrogate marker for the dose in the trigonum [30], which is the most sensitive region of the bladder, in our institutional protocols, patients are simulated and treated with a bladder filling of from 100 to 200 mL of saline. Thus, the D_2cc_ volume could be drawn further away from the trigonum (Figure 2). The threshold of the soft constraint was not met in two patients (1 and 4 in Table 3) with a very unfavorable anatomy and large initial tumors. The implant geometry was also suboptimal, which is the most important factor for achieving proper dose distributions [6]. While the first patient’s imperfect applicator placement could be partly attributed to inexperience, both patients had a narrow vagina with fibrotic vaginal vaults, which prevented the proper insertion of the colpostats and ovoids. Moreover, a dedicated ultrasound unit was unavailable in the department during the period in which the first four patients were treated. Ultrasound guidance during tandem insertion assisted in deciding on the proper depth as well as visual assessment of the position of the OARs and their proximity, especially in cases when additional repositioning of the applicator or changing the tandem angle was necessary. The utility of this method has been previously demonstrated and discussed [31,32].

Gastrointestinal toxicity is a major factor that has to be considered in the treatment of LACC. The first data on patients treated with intensity modulated radiotherapy with concurrent chemotherapy and IGABT were collected in the EMBRACE I study [33,34]. CTCAE version 3 was used to score the toxicity at that time. It was noted that D_2cc_ > 65 Gy doubles the incidence of proctitis. D_2cc_ > 75 Gy increases significantly the chance of fistula from 0% to 12.5% as well as the incidence of G2–4 toxicity up to 26%, while it is 10% in doses <69.5 Gy. D_2cc_ > 70 Gy in the rectum is also associated with late diarrhea. The sigmoid bowel D_2cc_ dose was successfully related to the severity of sigmoido-vaginal fistulas, as doses of 70–79 Gy were observed in patients with G3 fistulas, while those with a dose <70 Gy had G1–G2 fistulas or none [35]. Thus, the soft constraints for all bowel structures are D_2cc_ < 75 Gy, while the hard constraints are <65 Gy and <75 Gy for the rectum and sigmoid, respectively. In this study, the hard constraints were met for 88% of the patients. The first one, who had received high bowel doses, presented with rectal bleeding at the ninth month after completion of the therapy. Rectosigmoidoscopy revealed a small zone of inflammation in the upper rectum, and symptoms vanished after dietary adjustments. The second patient had the same symptom, and a telangiectasia was found. She was treated successfully with Etamsylate tablets apart from diet and probiotic intake. Both patients have reached two years of follow-up and have had their bleeding resolved.

The proposed protocol demonstrated safety and is subject to further improvements after the analysis of the initial results. Concurrent immune checkpoint inhibitors, higher doses in the nodal boosts, more precise evaluation of the risk factors for local and systemic failure, and further efforts to reduce the OTT will all come into consideration as we enroll more patients in this study. The low toxicity is encouraging to treat larger tumors, provided that good implant geometry can be achieved with the assistance of ultrasound guidance. Currently, the EBT system does not provide interstitial needles. In a country of limited resources, isotope-based radiotherapy can be a challenge as it requires a shielded bunker and changing the source over a certain period of time. The X-ray source, on the other hand, has no decay and is used only when a treatment session is necessary.

## 5. Conclusions

Electronic brachytherapy provides the option to deliver a high dose of radiation to the uterine cervix without leading to severe toxicity. Further actions must be undertaken to improve the local and systemic control in patients with LACC, which will lead to changes in the institutional protocol. Future studies with larger cohorts and longer follow-up are of crucial value in defining the proper patient selection for this treatment method.

## Figures and Tables

**Figure 1 cancers-17-02286-f001:**
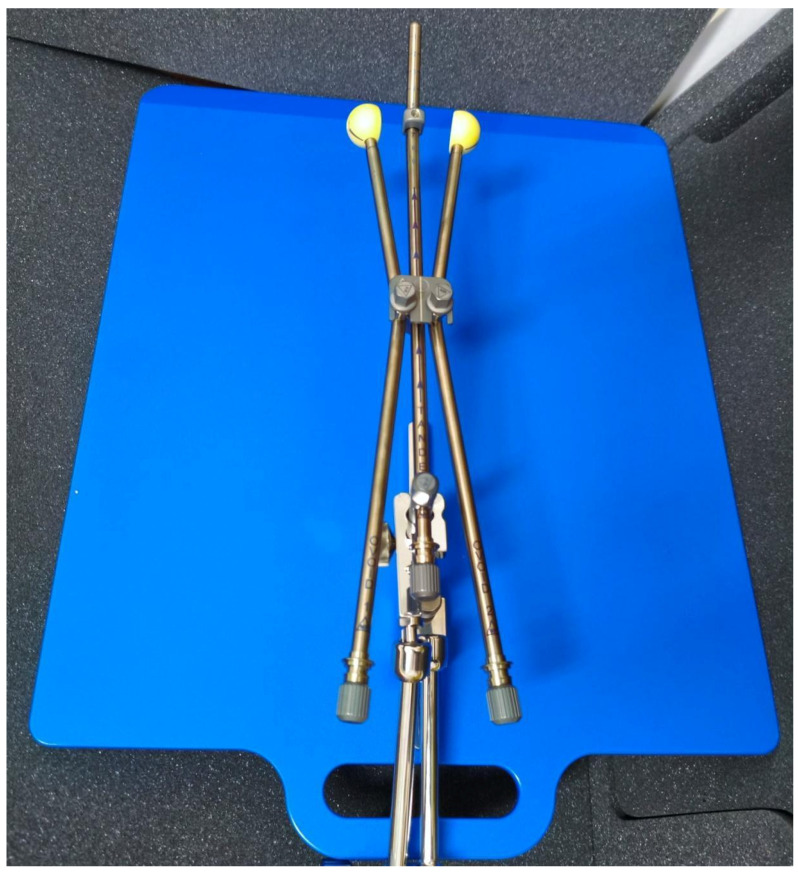
An assembled set of a tandem, colpostats, and ovoids locked to a base plate.

**Figure 2 cancers-17-02286-f002:**
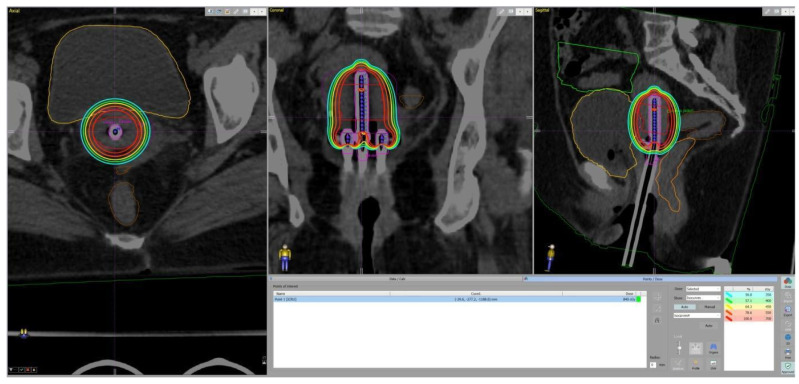
Dose distribution from the TPS. The isocurves represent the following doses: red—the prescribed dose; orange—the bladder D_2cc_ constraint; cyan—50% of the prescribed dose.

**Table 1 cancers-17-02286-t001:** Patients’ tumor stage and histology distribution.

Patient	Histology	TNM Stage	FIGO 2018 Stage	Number of Lymph Node Metastases	Short-Axis Diameter of the Largest Node (cm)
1	KSCC G2	T4N1	IVA	4	1
2	NKSCC G2	T4N0	IVA	0	
3	KSCC G2	T2aN0	IIA	0	
4	NKSCC G2	T2bN1	IIIC1	6	2
5	NKSCC G2	T1b3N0	IB3	0	
6	KSCC G2	T3bN0	IIIB	0	
7	NKSCC G2	T3bN0	IIIB	0	
8	NKSCC G2	T2bN1	IIIC1	8	2.8
9	NKSCC G2	T2bN0	IIB	0	
10	NKSCC G2	T3bN1	IIIC2r	11	2.8
11	NKSCC G2	T2bN1	IIIC1	2	1
12	NKSCC G2	T3bN1	IIIC2r	8	1.9
13	KSCC G2	T2bN0	IIB	0	
14	KSCC G2	T2bN1	IIIC1	2	0.8
15	KSCC G2	T2bN0	IIB	4	1.4
16	KSCC G2	T2aN1	IIIC1	3	2.2
17	NKSCC G3	T2bN1	IIIC1	2	2
18	NKSCC G2	T2bN0	IIB	0	
19	NKSCC G2	T2bN1	IIIC1	2	0.9
20	NKSCC G2	T2bN0	IIB	0	
21	NKSCC G3	T2bN0	IIB	0	
22	NKSCC G3	T2aN0	IIA	0	
23	STCC	T2bN0	IIB	0	
24	NKSCC G3	T4N0	IVA	0	
25	NKSCC G2	T2aN1	IIIC1	5	1.5

KSCC—keratinizing squamous cell carcinoma, NKSCC—non-keratinizing squamous cell carcinoma, STCC—squamotransitional cell carcinoma.

**Table 2 cancers-17-02286-t002:** Structures and planning aims. D90% and D95%—percentage of the target volume receiving the prescribed dose, and D_2cc_—maximum doses received in 2 cm^3^ of the contoured organ volume.

Structure	Dose Constraint
CTV HR	D_90%_ > 700 cGy
CTV HR	D_95%_ > 585 cGy
CTV HR	D_90%_ < 840 cGy
Bladder	D_2cc_ ≤ 550 cGy
Rectum	D_2cc_ ≤ 400 cGy
Sigmoid	D_2cc_ ≤ 450 cGy

**Table 3 cancers-17-02286-t003:** Dose distribution in the target volume and organs at risk for each patient.

n	CTV HR V_700cGy_ (%)	CTV HR D90 EQD_2_ (Gy)	Bladder D_2cc_ (Gy)	Bladder D_2cc_ EQD_2_ (Gy)	Rectum D_2cc_ (Gy)	Rectum D_2cc_ EQD_2_ (Gy)	Sigmoid D_2cc_ (Gy)	Sigmoid D_2cc_ EQD_2_ (Gy)
1	90	86	6.20	92.8	8.01	115.9	7.03	102.7
2	92	89	5.23	81.2	3.13	62.6	4.52	73.2
3	93	89	4.95	77.8	1.89	53.9	4.11	68.4
4	88	86	9.78	147.9	2.83	59.4	2.68	57.2
5	91	88	5.62	85.0	3.03	60.2	4.29	70.3
6	93	89	3.30	61.6	4.91	76.9	3.64	63.9
7	95	91	2.88	59.9	3.85	67.8	3.73	66.7
8	96	92	4.29	71.1	2.66	58.6	2.51	58.5
9	93	89	3.91	68.3	1.88	51.6	4.99	76.6
10	96	91	4.10	71.9	2.80	58.8	2.55	57.1
11	95	91	3.68	66.5	1.55	51.6	2.32	56.4
12	94	90	2.97	59.8	2.30	55.6	3.14	55.7
13	97	92	2.39	56.1	7.5	48.4	2.52	56.0
14	93	88	5.34	82.4	1.29	48.7	1.71	51.2
15	93	89	5.14	80.0	1.16	49.6	1.57	51.3
16	95	90	3.71	66.1	0.58	46.1	2.32	55.9
17	94	90	4.13	70.6	1.50	51.9	1.04	47.9
18	95	91	3.95	69.5	1.82	53.8	4.03	68.0
19	97	93	3.22	62.2	0.65	47.8	2.03	48.7
20	94	89	4.26	71.4	1.14	48.8	2.42	52.9
21	97	93	3.51	64.4	1.89	51.8	2.69	51.0
22	95	91	2.45	57.7	0.39	46.1	1.48	46.2
23	97	93	2.85	60.2	2.04	53.9	2.03	41.0
24	93	89	5.05	80.7	2.63	59.3	2.96	61.6
25	96	91	4.43	72.0	2.11	53.9	1.89	51.5

n—patient number in chronological order, CTV HR V_700cGy_—mean percentage of CTV HR volume receiving 100% of the prescribed dose, D90—cumulative dose delivered to 90% of the target volume, D_2cc_—mean values of the maximum doses in OARs received in 2 cm^3^ volume per fraction, EQD_2_—equivalent dose in 2 Gy fractions, accumulated from the whole treatment course.

**Table 4 cancers-17-02286-t004:** Patients with events of local, locoregional, or systemic failure.

Patient Number	Event	Time to Event Detection	Treatment	Outcome
4	boosted lymph node persistence	3 months	chemotherapy, stereotactic radiotherapy	partial response, developed bone metastasis 18 months after treatment
7	local failure, liver metastases	3 months	chemotherapy, stereotactic radiotherapy, immunotherapy	complete remission
8	lower third vaginal recurrence, cervical recurrence	18 months	chemotherapy	stable disease
10	local failure, lymph node metastases outside the treatment field, lung metastases	3 months	refused any treatment	death 20 months after treatment completion
11	local failure, lymph node metastases outside the treatment field	2 months	refused immediate treatment	death 13.5 months after completion of treatment
12	lymph node metastases outside the treatment field	9 months	chemotherapy, immunotherapy	complete remission
16	boosted lymph node persistence	3 months	stereotactic radiotherapy, surgery	partial response

## Data Availability

All available data can be provided by the corresponding author on demand.

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
