# Peer review of "Preliminary Experience with Electronic Brachytherapy in the Treatment of Locally Advanced Cervical Carcinoma"

_cancers, 2025, doi:10.3390/cancers17142286_

Round 1
Reviewer 1 Report
Comments and Suggestions for Authors
This study reports the dosimetric and clinical results of the treatment of locally advanced cervical cancer with electronic brachytherapy. In general, the work of this paper is clear and logical, but some points are required to be modified.
1.The sample size is insufficient, and it is recommended that the authors expand the cohort to enhance the statistical power and generalizability of the findings.
2.Please supplement the sample size calculation procedure.
Comments on the Quality of English Language1.“Patient’s tumor stage” should be “Patients’ tumor stage” in the title of Table 1.
2.The manuscript exhibits excessive use of passive voice constructions,so it is recommended to revise these into active voice for improved clarity and readability. For example, line 133,“The procedure was preceded by completing the steps of the quality assurance program...” can be modified to “Prior to the treatment procedure, our team completed the steps of the quality assurance program ...”
Author Response
Dear reviewer, thank you for finding to time to read and review this manuscript. Your comments are greatly appreciated.
Comments 1: The sample size is insufficient, and it is recommended that the authors expand the cohort to enhance the statistical power and generalizability of the findings.
Response 1: There are some changes we made in the manuscript to improve the clarity of the main conclusion- that the treatment protocol must and will change after the analysis of these results. The manuscript highlights the positives and negatives of following the EMBRACE protocol without making adjustments for changing the brachytherapy technique, as EBT delivers no dose to the pelvic lymph nodes for example. As it justifies the upcoming changes, it also provides some crucial data for other EBT users. We think that the medical community needs data urgently as EBT is commercially available without any guidelines for its use and there is a single study with only eight patients that was published a few years ago which guides us on the use of EBT for LACC. Last but not least, in the context of limited resources, this study was funded for the treatment of a limited number of patients and it was important to optimize the protocol as early as possible. The cohort is unlikely to expand significantly not only because we are changing the protocol but also because as now guidelines are updated and immunotherapy must be offered as a concurrent treatment for LACC, it would be unethical to decline these patients the opportunity to receive this treatment. In the and, our next patients will be treated differently. I hope that clarifies why this cohort is not expanding but the information about the results are still very valuable.
Comments 2: Please supplement the sample size calculation procedure.
Response 2: Thank you for pointing this out. We have not calculated a sample size as we are not comparing directly this cohort to any other except for the fact that we are trying to follow the EMBRACE guideline as strictly as possible. Bear in mind that at the point we started performing EBT for LACC, there was not a single hospital in our country that did image-guided adaptive brachytherapy because of the lack of resources but there is no official published data.
For the comments of the quality of English, we did our best to reduce the use of passive voice. Please , find the edited manuscript as an attached file. Thank you very much for your insights.

Reviewer 2 Report
Comments and Suggestions for Authors
Hi, thank you for providing me an oppturnity to review this manuscript! My comments are as follows.
- The sample size is too small
- I would like the author to address its limitations over the traditionally used chemotherapy
- Moreover i would like to know how easy is to start the setup of Electronic brachytherapy as compare to chemotherapy. Is it cost effective?
- What are its potential negative outcomes for long time use as according to this study this procedure has been carried out for less than 60 days.
Author Response
Dear reviewer, thank you for sparing the time to read and comment on the manuscript.
Comment 1: The sample size is too small
Response 1: As it might not have been clearly explained enough, we have made some changes in the manuscript purpose, discussion (line 326), and conclusion (line 338) to make it more clear that based on these results, there will be changes in the protocol, so this sample will not be comparable to the next enrolled patients. The only other previous study which treated LACC with EBT had eight patients. As the EBT system is now commercially available, it is critical for the medical community to have information on the the application of this method as soon as possible. Moreover, the funding of the study is also limited to treating a certain number of patients, as we are talking about limited resources in the article.
Comments 2. I would like the author to address its limitations over the traditionally used chemotherapy
Response 2: The chemotherapy used in this study is traditional according to the current European and American guidelines.
Comments 3: Moreover i would like to know how easy is to start the setup of Electronic brachytherapy as compare to chemotherapy. Is it cost effective?
Response 3: thank you for pointing cost-effectiveness out. We have added a few words on the cost of EBT - lines 330-333. As a radiation therapy method, it does not replace chemotherapy where it is needed and indicated. Consequently, there is no comparison needed, as radiation therapy (external beam plus brachytherapy) combined with chemotherapy are the current standard of care according to the guidelines.
Comments 4: What are its potential negative outcomes for long time use as according to this study this procedure has been carried out for less than 60 days.
Response 4: the manuscript discusses this point from lines 230 to 247. In case you emply dose intensification, the negative effects are desribed in lines 280-316.

Reviewer 3 Report
Comments and Suggestions for Authors
Dear Authors,
Yours start for prescribing sufficient doses to your patients is appreciable, but I think you rushed for publishing. You cannot be sure about the results and follow on the conclusions after just 1 year follow up.
It can be more reasonable maybe 2-3 years later.
Of course, your suggestion to use Electronic Brachytherapy for completing the cervical cancers treatment is very interesting specially for low-income countries. I hope we can have smaller size of X-Ray tubes in electronic interventional radiotherapy to use them even in interstitial needles.
You can change your approach to just introducing your devices and feasibility of them.
Author Response
Comments 1 :
Yours start for prescribing sufficient doses to your patients is appreciable, but I think you rushed for publishing. You cannot be sure about the results and follow on the conclusions after just 1 year follow up.
It can be more reasonable maybe 2-3 years later.
Of course, your suggestion to use Electronic Brachytherapy for completing the cervical cancers treatment is very interesting specially for low-income countries. I hope we can have smaller size of X-Ray tubes in electronic interventional radiotherapy to use them even in interstitial needles.
You can change your approach to just introducing your devices and feasibility of them.
Response 1: Dear reviewer, thank you for finding to time to read and review this manuscript. Your comments are greatly appreciated. There are some changes we made in the manuscript to improve the clarity of the main conclusion- that the treatment protocol must and will change after the analysis of these results. The manuscript highlights the positives and negatives of following the EMBRACE protocol without making adjustments for changing the brachytherapy technique, as EBT delivers no dose to the pelvic lymph nodes for example. As it justifies the upcoming changes, it also provides some crucial data for other EBT users. We think that the medical community needs data urgently as EBT is commercially available without any guidelines for its use and there is a single study with only eight patients that was published a few years ago which guides us on the use of EBT for LACC. Last but not least, in the context of limited resources, this study was funded for the treatment of a limited number of patients and it was important to optimize the protocol as early as possible.
We hope this answer and the updated manuscript are satisfying for you.
Round 2
Reviewer 3 Report
Comments and Suggestions for Authors
I satisfied after your report and reasons which you mentioned.